# The Bioaccessibility of Grape-Derived Phenolic Compounds: An Overview

**DOI:** 10.3390/foods14040607

**Published:** 2025-02-12

**Authors:** Danijel D. Milinčić, Nemanja S. Stanisavljević, Milica M. Pešić, Aleksandar Ž. Kostić, Slađana P. Stanojević, Mirjana B. Pešić

**Affiliations:** 1Institute of Food Technology and Biochemistry, Faculty of Agriculture, University of Belgrade, Nemanjina 6, 11080 Belgrade, Serbia; danijel.milincic@agrif.bg.ac.rs (D.D.M.); mpesic1801@gmail.com (M.M.P.); akostic@agrif.bg.ac.rs (A.Ž.K.); sladjas@agrif.bg.ac.rs (S.P.S.); 2Institute of Molecular Genetics and Genetic Engineering, University of Belgrade, P.O. Box 23, 11010 Belgrade, Serbia; nstan86@gmail.com

**Keywords:** phenolic compounds, bioaccessibility, digestion, grape, food matrices

## Abstract

Grape-derived phenolic compounds possess many health benefits, but their biological effectiveness and their effects on human health depend directly on bioaccessibility. Different physiological conditions, interactions with food compounds (proteins, lipids, and carbohydrates), and/or microbial transformations affect the solubilization and stability of phenolic compounds, thus altering their bioaccessibility and biological activity. Previously published review articles on grape-derived phenolic compounds have focused on characterization, transformation during winemaking, various applications, and health benefits, but the literature lacks a comprehensive overview of the bioaccessibility of these compounds during gastrointestinal digestion. In this context, models of gastrointestinal digestion and factors affecting the bioaccessibility of phenolic compounds were considered to understand the behavior of grape-derived phenolic compounds during digestion in the absence or presence of different food matrices. Finally, this review should enable the development of novel food products with targeted bioaccessibility of grape-derived phenolic compounds.

## 1. Introduction

Grapes are the most widely and intensively cultivated crop in the world (7.3 million hectares), with a global production of 80.1 million tons according to the 2022 census (OIV, 2022). Previous statistics show that the majority of grapes are used in the winemaking industry [1], while the rest are used for consumption (raw or dried), juice production, or the formulation of other food products [2,3]. Due to its great popularity worldwide, numerous studies have analyzed the bioactive compounds of grapes, wine, and winemaking by-products, with a special focus on phenolic compounds. It has been shown that grape phenolic compounds have a wide range of biological activities, such as antioxidant, antimicrobial, anticancer, anti-inflammatory and antidiabetic activities, as well as hepatoprotective, cardioprotective, and neuroprotective effects, which have previously been well-reviewed [4,5,6,7,8]. The well-known phenomenon of the “French paradox” best illustrates the positive influence of phenolic compounds from wine and their potent cardioprotective effects [7,9]. Furthermore, the development of healthier and sustainable food through the utilization of winemaking by-products as new food ingredients or extracts is a current trend in food innovation [10,11]. Numerous studies are focused on the extraction, valorization, and application of phenolic compounds from grape pomace [12,13,14,15,16,17,18,19] as one of the innovative approaches for waste utilization. In this regard, various value-added food products (bakery, confectionary, and dairy products) enriched with grape-derived phenolics compounds (extracts or powders) have been developed and characterized, as evidenced by numerous review papers published in previous years [6,8,13,14,20,21].

However, the therapeutic and biological properties of phenolic compounds from grapes, wine, and winemaking by-products are closely linked to their metabolism as these compounds undergo various biochemical transformations before absorption [4]. Their application is often limited by their low solubility and stability, while their bioaccessibility directly depends on the amount of phenolic compounds released from the complex food matrix during digestion. Food components such as proteins, carbohydrates, and lipids have been found to significantly affect the bioaccessibility of phenolic compounds, as well as their digestibility and antioxidant activity due to interactions between them [15,22,23,24,25,26]. Therefore, the bioaccessibility of phenolic compounds during and after gastrointestinal digestion is a crucial step for their effects on human health.

Some previous review articles have addressed the bioaccessibility and bioavailability of phenolic compounds in pumpkins [27], seaweed [28], bread [29], cereal grains [30,31], tropical fruits [32], etc. Grapes and wine have been extensively studied in recent years, with a focus on the winemaking process, waste valorization, and their further use in the food industry, as evidenced by numerous reviews. The behavior of grape-derived phenolic compounds during gastrointestinal digestion has been increasingly investigated via static in vitro digestion models and, less frequently, in vivo studies to gain a better insight into their bioaccessibility and functionality. To our knowledge, there is no review article summarizing all of the research on the bioaccessibility of grape-derived phenolic compounds, which would allow for better reproducibility/comparability of the results obtained on this topic. On the other hand, the bioaccessibility of grape-derived phenolic compounds in the presence of different food matrices has hardly been investigated. Only a few studies analyze the bioaccessibility of grape-derived phenolic compounds in the presence of complex matrices such as dairy products, egg products, infant formula, or bakery products [33,34,35,36,37].

Considering all the review articles on grape-derived phenolics published so far, a comprehensive and critical overview of the bioaccessibility of grape-derived phenolic compounds during gastrointestinal digestion is lacking in the literature. This review should enable researchers to understand the behavior of grape-derived phenolic compounds during gastrointestinal digestion in the absence or presence of different food matrices, allowing for the development of novel food products, natural additives, or dietary supplements with the targeted bioaccessibility of grape-derived phenolic compounds to achieve the best effect on human health.

## 2. Methodology for Literature Review

This literature review was conducted according to the methodology proposed by Wee and Banister [38]. It consists of the following stages: selection of articles from the SCOPUS database using keywords (1453); screening of the titles of selected articles and exclusion of inappropriate articles (691); further screening of the remaining articles using the abstract (292); and the final selection was conducted by screening the full text of the articles to obtain the articles included in the text (136). The procedure was repeated to add 30 new references after the revision of this manuscript requested by the reviewers.

## 3. Grape, Wine, and Grape Pomace as Sources of Phenolic Compounds

Grapes are an important source of phenolic compounds, which are manly found in the skin (28–35%), seeds (60–70%), and stems, while their content in the pulp is much lower and does not exceed 10% [11,39,40]. During winemaking, phenolic compounds are extracted from the grapes and transferred to the must and wine, but a considerable amount of these compounds remains in the grape pomace [41]. It is estimated that about 70% of the total phenolic compounds remain in the grape pomace after processing, while the majority of extractable phenolic compounds originate from the seeds, which account for 38–52% of the winemaking by-products [1]. The seeds contain a considerable amount of flavan-3-ols, pro(antho)cyanidins, and phenolic acids [11,42,43]; the skins contain mainly flavonols and anthocyanins [40,44,45]; while the stems contain mainly stilbenes and pro(antho)cyanidins [46] (Figure 1). The skin of white grape varieties contains significantly fewer phenolic compounds than the skin of black grape varieties, as it does not contain anthocyanins. In recent years, the application of advanced HPLC-MS techniques in the analysis of grape, wine, and grape pomace samples has contributed to an improved identification and structural characterization of individual phenolic compounds. The use of high-resolution mass detectors such as Orbitrap or Q-ToF enables the identification and structural elucidation of new (unknown) phenolic compounds in samples of grape-derived products [11,42,44,45,47,48,49,50,51].

### 3.1. Phenolic Acids

The phenolic acids detected in grape seeds, skins, stems, wine, and pomace are usually present in free form or conjugated with sugar units and/or tartaric acid [52]. Gallic acid is the most frequently quantified phenolic acid in the seeds of different grape varieties [40,53,54,55], while ethyl and methyl gallate have been detected in wine, as well as in pomace and seeds analyzed after fermentation [56]. Ellagic acid (a dimeric derivative of gallic acid) was confirmed in grape seeds, pomace, and wine [39,57,58]. A high concentration of ellagic acid was found in the seeds of the indigenous Prokupac grape variety [39,40] and its clones [43]. The presence of ellagic acid and ellagitanins in wine may be an indicator of wine ageing and extraction from oak barrels during maturation [52,59]. Hydroxycinnamic acids (coumaric, caffeic, and ferulic acids) are most commonly found in grape skins and wine in conjugation with tartaric acid as *cis/trans* isomers of feruloyltartaric, caffeoyltartaric, and coumaroyltartaric acid [11,42,44,45,60].

### 3.2. Stilbenes

Stilbenes are typical compounds of grapes and wine that are widely known for their health benefits. Numerous in vitro and in vivo studies have shown various health benefits of resveratrol, such as antiageing, anticarcinogenic, antidiabetic, anti-inflammatory, antioxidant, and cardioprotective properties [61,62]. However, resveratrol has low solubility and stability, which often limits its bioavailability and activity. Resveratrol and its glycoside piceid have been found in grape pomace, seeds, skins, stems, and wine [42,44,46,63]. *Cis*-resveratrol was found most frequently in wine and less frequently in grapes [52]. The content of resveratrol in wine is often significantly lower compared to other phenolics, but has attracted attention due to its good therapeutic properties [61]. In addition, the content of resveratrol and other stilbenes in wine and grape juice is highly dependent on processing and storage [62]. Dimers, trimers, and tetramers of resveratrol have been identified in grape seeds and skins [11,45]. In addition, oligomeric oxidation forms of resveratrol, known as viniferins (mainly *Ɛ*-viniferin), are predominantly present in grape stems [46,64,65,66].

**Figure 1 foods-14-00607-f001:**
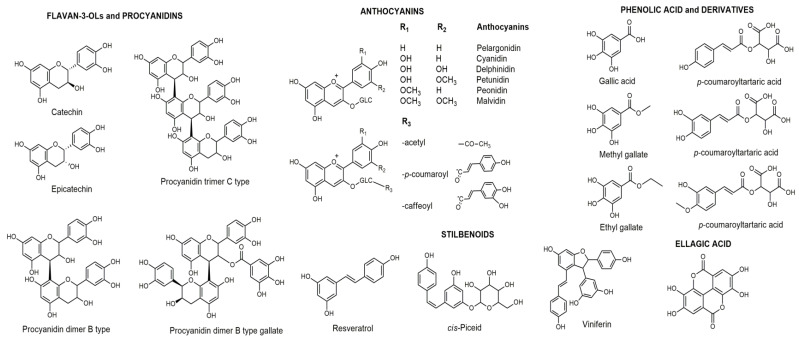
Main phenolic compounds present in grapes, wine, and by-products.

### 3.3. Flavan-3-Ols and Procyanidins

Flavonoids in grape pomace, skins, and seeds are mostly present in the form of glycosides, which form *O*-glycosides with monosaccharides or disaccharides. According to the literature, the seeds contain a significantly higher concentration of flavonoids than the skin [40,42]. Monomers, dimers, oligomers, and polymers of flavan-3-ols are the predominant phenolic compounds in the grape seeds, but they can be also found in the skin, stems, wine, and pomace. The most important representatives of flavan-3-ols in grapes and wine are catechin, epicatechin, gallocatechin, epigallocatechin, and their gallates [40,67]. Dimers, trimers, and tetramers of the procyanidin B-type, as well as their gallates, were manly confirmed in grape seeds, followed by wine [11,39,44,51,60,68].

### 3.4. Flavonols

Flavonols are predominantly found in grape skin and wine. The flavonols most frequently identified in grapes and wine are quercetin, myricetin, isorhamnetin, syringetin, laricitrin, kaempferol, taxifolin, and their glycosides, diglucosides and hexuronides [56,58,67]. Flavonols are localized in the skin mainly as 3-O-glycosides or, more rarely, as 7-O-glycosides or aglycones. Quercetin and its derivatives have been found predominantly as flavonols in the skin of both international (Merlot, Cabernet Sauvignon, Riesling, Sangiovese) and indigenous grape varieties (Smederevka, Prokupac, and Crna Tamjanika) [40,41,53,69,70,71].

### 3.5. Anthocyanins

Anthocyanidins are unstable molecules that rarely occur in nature in the form of aglycones. They are mainly found in the form of glycosides (anthocyanins) and are responsible for the color of the grape skin, particularly the malvidin derivatives, which are dominant in the skins of the black grape varieties of the species *Vitis vinifera* [72]. The predominant presence of malvidin-3-*O*-glucoside, petunidin-3-*O*-glucoside, and peonidin-3-*O*-glucoside has been confirmed in grape varieties such as Cabernet Sauvignon, Merlot, Muscat Hamburg, and Prokupac [73,74,75], as well as in red wine [45,59,76]. In addition, various acetyl, caffeoyl, and coumaroyl derivatives of anthocyanins, as well as pyranoanthocyanins, which are formed via the condensation of anthocyanins with pyruvate and acetaldehyde (vitisin A and B) during wine aging, can frequently be found in the grape skin [40,44,45,56,72,77]. Other flavonoids (flavanones and flavones) are not specific to grapes and wine, but naringenin, hesperetin, eriodictyol, apigenin, luteolin, and their glycosides have been detected in small amounts in grape seeds, skins, wine or pomace [40,49,58].

## 4. Extraction and Stabilization of Grape-Derived Phenolic Compounds

Extraction, stabilization, characterization, and quantification are important steps in the use of grape phenolic extracts obtained from different sources (primarily pomace and its constituents) [12,13,78]. Extraction techniques have been intensively researched in recent years to obtain a sufficient amount of phenolic compounds for their potential commercialization [79]. The choice of extraction method depends on the qualitative and quantitative composition of the extract, the plant material, and the physicochemical properties of the dominant phenolics to be extracted, such as solubility, polarity, hydrophobicity, and thermal stability [2,80]. There is no standard extraction method, but solid–liquid extraction is most commonly used to extract phenolic compounds from pomace, seeds, and/or skins, followed by mechanical stirring or ultrasonic treatment [2,12,78,81]. Solid–liquid extraction is based on the contact of a solid matrix with an extraction solvent, which leads to the migration of dissolved molecules (mass transport) from the solid [2,12]. The efficiency of the extraction and stabilization of phenolic compounds can be controlled by changing the concentration gradient, diffusion coefficient, or boundary layer, or by choosing appropriate experimental parameters such as temperature, extraction time, pH, solid solvent ratio, particle size, mixing, and solvent polarity [2,12,80,82]. Methanol, ethanol, acetone, and water are most commonly used for extraction of phenolic compounds from grape pomace [82,83]. However, the efficiency of extraction of phenolic compounds from seeds and skins is better in polar aqueous–alcohol solutions (aqueous solutions of methanol or ethanol) and aqueous acetone solutions due to the polar nature of these molecules [84,85]. Thus, Spigno et al. [83] showed that the concentration of phenolic compounds in the “Barbera” skin extracts increased by using different aqueous/ethanol mixtures (from 10% to 60%). Ethanol is an accessible and inexpensive extraction solvent, has GRAS status, and is preferable for later use of the extract in the food industry [12]. On the other hand, methanol (100%, 80%, or 50% methanol) has been shown to be the best extraction solvent for anthocyanins from pomace [58] and monomeric flavan-3-ols from seeds [84]. However, extracts for further use in the food industry often require additional processing, evaporation in a vacuum evaporator to remove the alcoholic extractant, and final reconstitution in water. The extraction time, temperature, and pH of the extractant are very important parameters that influence the stability, utilization, and efficiency of extracting phenolic compounds [12,86]. It has been shown that extraction temperature of 60 °C and the use of “acidified” extraction media (usually containing 0.1 to 1% HCl) increase the content and stability of total phenolic compounds and anthocyanins extracted from grape skin [86]. To obtain extracts with the highest content of phenolic compounds, extraction optimization is often performed using the response surface methodology [87,88]. In recent years, the microwave-assisted method, pulsed electric fields, high-voltage electric discharges, pressurized liquid, the enzyme-assisted method, and hydrothermal methods have been applied to produce high-quality and safe extracts [2,12,78,81,89]. However, these techniques are still under development or applied at the laboratory level and require high maintenance costs, specialized personnel, and additional studies [2,12] (Coelho et al., 2020; Fontana et al., 2013).

## 5. Factors Influencing the Bioaccessibility/Bioavailability of Phenolic Compounds

Phenolic compounds are an integral part of daily meals in the human diet and understanding their bioaccessibility/bioavailability represents a crucial step in assessing their biopotential [90]. The bioaccessible fraction of phenolic compounds is defined as a set of molecules released from the matrix in the gastrointestinal tract that are available for intestinal absorption and biotransformation by the gut microbiota, and which potentially exhibit bioactivity [91,92,93,94]. Therefore, the bioaccessibility presents a key parameter for evaluating the functional efficiency of bioactive compounds, as the body often does not utilize all nutrients and bioactive components from food [95]. The bioaccessibility of phenolic compounds mostly depends on the characteristics of the food matrices (fruits, vegetables, or complex food products), the physicochemical properties of phenolic molecules, and the physiological conditions encountered in different phases of digestion (including enzyme concentration and pH). For these reasons, various in vitro digestion models have been developed that simulate human physiological conditions to predict the bioaccessibility of phenolic compounds in different food matrices. However, phenolic compounds are mostly present in the form of glycosides, esters, or polymers, and their absorption depends on the activity of enzymes present in the stomach and intestine, as well as on their conversion into metabolites caused by the gut microbiota. Therefore, bioavailability also depends on the ability of phenolic compounds to cross the gastrointestinal barriers and enter the bloodstream; i.e., it represents the link between the bioaccessibility and the bioactivity of phenolic compounds and/or their metabolites. Thus, bioavailability can be defined as the amount of phenolic compounds that are fully digested and absorbed and exhibit the target bioactivity in the human body [94]. In this case, the absorption of phenolic compounds depends on their interaction with enzymes/salivary proteins, their conversion by digestive enzymes, the pH effect, and the effect of the gut/colonic microbiota. Finally, only formed metabolites and intact polyphenols can be absorbed into the bloodstream and transported to the liver and target tissues, where they exert biological effects or are excreted in the urine or feces [90,94].

### 5.1. The Influence of Food Matrices

The interactions between the food matrix and phenolic compounds during in vitro digestion have been extensively studied over the last decade and are well described [15,22,96]. The bioaccessibility of phenolic compounds can be influenced by all food compounds, such as proteins, carbohydrates, lipids [22,26], or fibers [97,98], and it depends on the following: (1) the structure and solubility of phenolic compounds; (2) the composition, texture, granulation, and digestibility of the food matrix; (3) the interactions of phenolic compounds with macronutrients (proteins, carbohydrate, and lipids), their hydrolysates (peptides, fatty acids, and oligosaccharides), and digestive enzymes; and (4) the composition of the digest [22,23,33,96,99,100,101,102,103]. Phenolic compounds are often associated with macromolecules from the matrix through reversible interactions (hydrophobic interactions, hydrogen bonds, and van der Waals interactions) or, more rarely, through covalent bonds that can occur in the case of proteins [22,25,104]. These interactions contribute to the improved oxidative stability of phenolic compounds during gastrointestinal digestion. However, in most cases, the presence of the matrix limits the release of phenolic compounds in the digestive tract, reducing their bioaccessibility and ability to exert biological effects [22,102]. Phenolic compounds, for example, are most commonly bound to proteins by hydrophobic interactions and subsequently stabilized by hydrogen bonds, but other types of interactions, such as van der Waals and ionic interactions, can also occur [22]. The structural flexibility, the number of OH groups, and the molecular mass of phenolic compounds play an important role in their interactions with proteins [22]. The interactions between phenolic compounds from grapes, tea, coffee, cocoa, and milk proteins have been studied most frequently [25,26,37,105]. These interactions between phenolic compounds and dietary proteins may affect the following: (1) changes in protein structure; (2) the net charge of protein molecules; (3) reduction in the bioaccessibility of certain amino acid residues; (4) protein digestibility; (5) alteration of the functional and techno-functional properties of proteins; (6) reduction in the activity of digestive enzymes; (7) the bioaccessibility of phenolic compounds (reduction/increase); and (8) the modification (“masking”) of the antioxidant and biological properties of phenolic compounds [22,24,25,26,99]. Finally, numerous studies have shown that the structure, the composition of the food matrix, and the co-digestion effect of phenolic compounds with different food constituents influence not only their bioaccessibility but also digestibility and antioxidant activity [33,35,36,37,106].

### 5.2. The Influence of Physicochemical Properties of Phenolic Molecules

The hydrophilic/lipophilic balance is crucial in solubilizing hydrophilic phenolic compounds in the aqueous phase of intestinal digest and the reconstitution of lipophilic phenolic compounds to the mixed micelle phase [107]. Most phenolic compounds (with a few exceptions) are considered moderately to highly water-soluble, and their bioaccessibility primarily depends on the release from the matrix and the solubilization in the aqueous phase [107]. On the contrary, the bioaccessibility of lipophilic phenolic compounds depends on the efficient micellarization and presence of bile salts and pancreatic enzymes [107]. Furthermore, many factors, such as the release of phenolic compounds from the food matrix, low solubility or the formation of insoluble complexes in the digestive tract, interaction with salivary proteins, sensitivity to pH conditions, low permeability of the intestinal mucosa, and molecular transformations in the digestive tract can reduce the in vivo bioaccessibility of phenolic compounds [91,92,107,108].

### 5.3. The Influence of Physiological Conditions Encountered in Different Phases of Digestion

Digestion of food begins in the mouth and involves chewing and salivary secretion as two complementary oral mechanisms [109]. Chewing and the low pH of the stomach initiate the release of phenolic compounds from softened and disintegrated food matrices [110]. Thereafter, the phenolic compounds dissolved in the digestive fluid of the gastric and intestinal phases are further absorbed and metabolized. For example, the bioaccessibility of grape-derived phenolic compounds (anthocyanins and/or procyanidins) is often variable due to their tendency to form complexes with proteins and to react with the enzymes and constituents of the digestive cocktail [99,101,108,111]. Some studies have shown that there is a significant loss of catechin after in vitro digestion as it tends to interact with digestive enzymes, masking and limiting its detection [112,113]. In addition, the bioaccessibility and stability of phenolic compounds in the intestinal phase also depend on the environment pH [114]. For example, pure standards of caffeic acid, gallic acid, catechin, quercetin, and resveratrol were found to be stable under gastric conditions, while their content decreased by 24.9%, 43.3%, 7.2%, 5.8%, and 69.5%, respectively, after intestinal digestion in moderate alkaline conditions compared to the content in the initial solution [115]. In addition, Henning et al. [116] reported a decrease in the bioaccessibility/bioavailability of flavan-3-ols (contained in tea and grape seeds) but showed that non-gallated flavan-3-ols have better stability at pH 7 (intestinal pH) than gallated forms. Anthocyanins are often the most sensitive to pH changes during digestion. In grapes/wine, anthocyanins are present in the form of flavylium ions, which readily convert to carbinol pseudo-bases, quinoidal-bases, or chalcones as pH increases during digestion, directly affecting their solubility and bioaccessibility [117]. Most anthocyanin exhibit high stability under acidic gastric conditions, whereas they are unstable and easily degraded in neutral or weakly basic intestinal conditions. Acylation of anthocyanin contributes to increasing stability at higher pH [117]. Therefore, the quantification of anthocyanins after in vitro digestion is often complicated due to the pH-dependent equilibrium of their structural forms and co-pigmentation. Ellagitannins are also sensitive to acidic or basic environments as they can gradually hydrolyze to hexahydroxydiphenic acid, which further convert to ellagic acid [107]. It should be noted that antioxidant activity is also pH-dependent due to the deprotonation of hydroxyl groups present on the aromatic rings of phenolic compounds, which can give apparently higher or lower results for antioxidant capacity [115]. Di Majo et al. [118] showed that increasing the pH from 3.5 to 7.4 significantly increases the antioxidant activity of most flavonols, flavan-3-ols, and phenolic acids, which is due to the different degrees of dissociation of the substituted functional groups, mainly hydroxyl substituents. Bearing the above-mentioned limitations in mind, the bioaccessibility of phenolic compounds can be increased and modulated by encapsulating them in appropriate carriers resistant to gastrointestinal conditions, which release phenolic compounds at targeted sites in a controlled manner and enhance their absorption [91,96,119,120,121,122,123].

### 5.4. Biotransformation of Phenolic Compounds

Phenolic compounds are mostly present in the form of esters, glycosides, and polymers, which cannot be absorbed in vivo in these forms. Transformations of derivatives of phenolic compound under the influence of intestinal microflora and enzymes (β-glucosidases, β-rhamnosidases, and esterases) contribute to the formation of glucuronated, methylated, and sulfonated derivatives with altered functional properties [110,124]. Anthocyanins, for example, are only absorbed to a limited extent in the gastrointestinal tract as they are rapidly converted into absorbable metabolites (a few hours after consumption) [111]. The bioaccessibility and absorption of anthocyanins are influenced by their hydrophilic/hydrophobic nature and the number of hydroxyl groups and the sugar residue (mono- or oligosaccharide) in their structure. In the presence of gut microflora, the glycosidic bonds are usually cleaved, and the anthocyanidin rings are broken, except for anthocyanidins bound to pentoses and acids, which show increased resistance to degradation [111,114]. Oligomeric and polymeric compounds also undergo biotransformation, reducing their molecular mass and forming derivatives that are more easily absorbed [110]. Procyanidins are easily broken down under gastrointestinal conditions into their flavan-3-ol monomers (mainly catechin and epicatechin), which are further absorbed and metabolized [5]. Considering these specific transformations, the analysis of in vivo metabolites represents a major challenge for researchers.

## 6. Models for Investigation of Gastrointestinal Digestion of Grape-Derived Phenolic Compounds

Depending on the models used, studies on gastrointestinal digestion can generally be divided into two groups: in vitro (static and dynamic models) studies and in vivo studies. In vitro studies are more commonly used to determine the bioaccessibility of phenolic compounds as they are fast, safe, and have no ethical limitations compared to in vivo experiments. In vitro studies mainly simulate physiological conditions that occur during digestion in the human gastrointestinal tract [96]. In vitro studies are most commonly used for food analysis and usually involve three phases of the digestive system (mouth, stomach, and intestine). In view of these facts, special attention should be paid to the following parameters in in vitro experiments: temperature, mixing rate, duration, and enzymatic composition of saliva, gastric, and intestinal fluids (Table 1). In addition, cell lines (Caco-2 cells) that mimic the intestinal mucosa are used in advanced in vitro studies to investigate the active transport of bile acids and bioactive molecules (mainly phenolic compounds) [95,114]. Static in vitro models have been most commonly used to assess the bioaccessibility of phenolic compounds from grapes, wine, and pomace (skin, seed, and stem) (Table 1 and Figure 2).

On the other hand, dynamic in vitro models are more expensive and more demanding to perform and to interpret the results [98,109,125]. In addition, the static in vitro models used differ in their sophistication, precision, and application. The most commonly used standardized in vitro digestion model is proposed by the COST INFOGEST network [126,127]. The physiological relevance of the in vitro INFOGEST protocol was compared and evaluated with an in vivo pig experiment, focusing on the protein digestibility (skimmed milk powder) [126,128]. The results showed an overlay (similar patterns) between the peptide fractions of the harmonized INFOGEST protocol and the in vivo digestion at the end of the gastric and intestinal (jejunum) phases [128,129]. However, the assessment of the bioaccessibility of different phytochemicals (phenolic compounds, carotenoids, betalains, etc.) is a challenge for researchers, and the results obtained are often variable from laboratory to laboratory. Previous experience has shown that slight changes in digestive parameters can significantly affect the bioaccessibility and recovery of phytochemicals, especially phenolic compounds.

Various static in vitro digestion models can be found in the literature that use different experimental conditions, including the pH of the fluids, the mixing speed, the incubation time, and the composition of the digestive fluids, to simulate conditions in the mouth, stomach, and intestine [92]. The composition of the digestive fluids and the concentrations of the constituents (buffers, salt, and enzymes) can be variable and depend on the research objectives [130]. For all digestive phases, the temperature is constant and set at 37 °C, but the pH is variable and specific to each phase, namely, the oral phase (pH ~6–7), gastric phase (pH ~1–5), and intestinal phase (pH ~6–7) (Figure 2). Physiological conditions (e.g., the type and quantity of enzymes, salts, buffers, and emulsifiers) also vary between in vitro digestion models and have a significant effect on the repeatability, comparison, and final interpretation of the results [103,109,130]. The simulated oral phase in a laboratory model involves mixing extracts or matrices with simulated salivary fluid under specific conditions (5–10 min; T = 37 °C; and pH 6.5–6.8). The most important factors influencing the behavior of phenolic compounds in this phase are as follows: (1) interactions with saliva; (2) interactions with mucin; (3) interactions with the enzymes used in this phase (α-amylase) [96]. Previous research indicates the formation of insoluble complexes in the oral phase (perceived as astringency) due to the interaction of flavan-3-ols, proanthocyanidins, and some anthocyanins with salivary proteins [72,96,131,132,133]. It has also been shown that the chemical binding affinity of grape flavan-3-ols/procyanidins and salivary proteins is often strongly dependent on the concentration of phenolics/proteins and the degree of polymerization/galloylation of the monomeric flavan-3-ols [132]. In addition, recent studies have demonstrated the interaction between various grape anthocyanins (acetylated, coumaroylated, and cinnamoylated anthocyanins) and salivary proteins and their effects on mouthfeel properties [133,134]. After the oral phase, the “bolus” is mixed under specific conditions (1–2 h; T = 37 °C; and pH 1–5) with simulated gastric fluid and one or more enzymes e.g., porcine gastric pepsin (some models also contain gastric lipase). After the gastric phase, the “chyme” is mixed under specific conditions (2 h; T = 37 °C; and pH 6–8) with simulated intestinal fluid, usually containing bile salts and porcine pancreatin (a mixture of amylase, lipase, and proteinase). More complex models in this phase also contain small organic molecules, coenzymes, phospholipids, and phospholipase A2. Bile salts are an integral part of all in vitro digestion models and play an important role in lipid digestion and the solubilization of lipophilic phenolic compounds. After intestinal digestion, dialysis and centrifugation are commonly used to separate the bioaccessible fraction, which is then further analyzed [91,92,107,108,109,127]. Some in vitro models also include a colon phase, which contains a complex microbial ecosystem that ferments food components that could not be broken down in the upper gastrointestinal tract. Depending on their structure, phenolic compounds in the colonic phase are converted into metabolites such as phenolic aglycones, various phenolic glucuronide sulphate and/or methylated derivatives, urolithin, hydroxyphenylacetic acid, hydroxyphenylpropionic acid, and phenylvalerolactones, which are absorbed and exhibit potential bioactivities [91,93,107,108].

**Table 1 foods-14-00607-t001:** Bioaccessibility of total and/or selected grape-derived phenolic compounds after in vitro digestion in the stomach, intestine, and/or colon.

Sources	Phenolic Compounds	Gastric Phase (GF) Conditions	Bioaccessibility GF (%)	Intestinal Phase (IF) Conditions	Bioaccessibility IF (%) (Compared with Initial Sample)	Bioaccessibility After Colon Phase (%)	Observations/Remarks	Ref.
“Red Globe” grape (whole grape)	^1^ Total phenolics ^1^ Total flavonoids ^1^ Total anthocyanins	(a) Pepsin (300 U/mL) (b) T = 37 °C; pH 2.0; τ = 2 h	60.63% 48.19% 35.71%	(a) Pancreatin (0.8 g/L); bile salt (25 mg/mL) (b) T = 37 °C; pH 7.5; τ = 2 h (c) Digest (centrifugation)	62.38% 56.07% 7.63%	/	Antioxidant properties of digested samples (after IF): (a) ABTS^•+^ ↑ (b) FRAP ↑	[115]
Grape seed extract	^2^ Catechin ^2^ Epicatechin ^2^ Procyanidin B2 ^2^ Procyanidin B3	(a) Pepsin (800–1000 U/mg proteins) (b) T = 37 °C; pH 2.0; τ = 1 h; 55 rpm/min	98.3% 96.6% 109.3% 124.9%	(a) Pancreatin (2 g/L); bile salt extract (25 mg/mL) (b) Presence/absence of Caco-2 cells (c) T = 37 °C; pH 7.0; τ = 2 h	Presence/absence of Caco-2 cells 56.1/n.d% 14.7/n.d% n.d/n.d n.d/n.d	/	Recovery of catechin after intestinal digestion was increased from 59% to 98% (after extraction with acetonitrile, which “unmasks” catechin bound to digestive enzymes)	[113]
Red wine	^2^ Total phenolics ^2^ Total anthocyanins ^2^ Malvidin-3-O-glucoside	(a) Pepsin (315 U/mL) (b) T = 37 °C; pH 1.7; τ = 2 h; 100 rpm	66.4% 99.1% /	(a) Pancreatin (4 mg/mL); bile salt (25 mg/mL) (b) T = 37 °C; pH 7.5; τ = 2 h (c) Digest (dialysis)	After dialysis 7.2% 3.7% 0.2%	39.7% 34.1% 0.9%	/	[135]
Grape pomace (flour and extract)	^1^ Total phenolics	(a) “porcine” pepsin (b) T = 37 °C; pH 2.0; τ = 2 h (c) Digest (filtration; 5 kDa membrane)	Retentate/permeate (mg/g) 0.82/2.86% (flour) 58.22/35.23% (extract)	(a) “porcine” pancreatin (b) T = 37 °C; pH 7.0; τ = 2 h (c) Digest (filtration; 5 kDa membrane)	Retentate/permeate (mg/g) 24.16/5.76% (flour) 68.32/4.21% (extract)	/	Antioxidant properties of digest (compared to the initial sample) (a) TEAC (Retentate/permeate) ↑ (b) ORAC (Retentate/permeate) ↑ COST INFOGEST protocol of digestion	[136]
Grape juices (*Vitis vinifera* L.)	^1^ Total phenolics	(a) Pepsin (pepsin/juice; 1:10 w/w) (b) T = 37 °C; pH 2.0; τ = 1 h	/	(a) Trypsin (trypsin/juices; 1:10 w/w) (b) T = 37 °C; pH 6.0; τ = 2 h (c) Digest (centrifugation)	After intestinal digestion, the content of total phenolic compounds is twice as high	/	Antioxidant properties of digest (compared to the initial sample) (a) FRAP ↑ (b) α-glucosidase inhibitory activity ↑ (c) α-amylase inhibitory activity (no significant difference)	[137]
Fruit mix (with a portion of Airen grape concentrate)	^1^ Total phenolics	(a) Pepsin (0.02 g/g sample) (b) T = 37 °C; pH 2.0; τ = 2 h, 120 rpm/min	/	(a) Pancreatin (0.005 g/g sample); bile salt (0.03 g/g sample) (b) T = 37 °C; pH 6.5–7.2; τ = 2 h (c) Digest (centrifugation)	90.26% (Fruit mix) 91.51% (Fruit mix + Fe) 76.52% (Fruit mix + Zn) 76.52%(Fruit mix + Fe + Zn)	/	Antioxidant properties of digest (compared to the initial sample) (a) ORAC ↑ (for all sample) (b) ABTS^•+^ ↑ (for all sample)	[138]
Pomace (Merlot); Lyophilized extract	^2^ Total phenolics ^2^ Total gallic acid derivatives ^2^ Total flavan-3-ols and procyanidins ^2^ Catechin/Epicatechin ^2^ Total flavonols ^2^ Total anthocyanins ^2^ Peonidin/malvidin-3-O-glucoside	(a) Pepsin (0.32 g/100 mL) (b) T = 37 °C; pH 1.2; τ = 2 h; 150 rpm	/	(a) Pancreatin (0.15 g/100 mL); bile salt (0.9/100 mL) (a) T = 37 °C; pH 6.0; τ = 1 h; 150 rpm	12.85% 1.47% 12.1% 24.34/23.69% 15.46% 16.03% 14.59/9.98%	12.88% 1.49% 12.35% 19.80/20.96% 14.80% 15.16% 13.89/9.1%	Functional properties after colon fermentation (compared to the initial sample) (a) DPPH^•^ ↓ (b) Reduction properties ↑ (c) β-caroten “bleaching” inhibition ↓ (d) Cytotoxic properties (MCF-7; NCI-H460; HepG2) ↑ (e) Antimicrobial properties ↓ or no significant effect	[139]
Grape Wine (Syrah)	^2^ Total phenolics ^2^ Total phenolic acids ^2^ Total flavonols ^2^ Total flavan-3-ols and procyanidins ^2^ Catechin ^2^ Epicatechin ^2^ Total anthocyanins ^2^ Peonidin-3-glucoside ^2^ Malvidin-3-glucoside ^2^ Malvidin-3-acetil-glucoside	(a) Pepsin 450 U/g (mL) (40 mg/mL) (b) T = 37 °C; pH 2.0; τ = 2 h	Grape/Wine 68.55/127.47% 80.2/143.11% 73.11/129.65% 138.6/179.66% 325.5/160.42% 379.9/246.31% 45.07/100.82 31.6/125.67% 55.5/116.32% 56.36/61.72%	(a) Pancreatin (1.2 mg/g(mL)); bile salt (5.6 mg/g(mL)) (b) T = 37 °C; pH 7.5; τ = 2 h (c) Digest (dialysis)	Grape No-D(D)/Wine No-D(D) 4.71(7.94)/6.31(4.82)% 24.7(60.2)/13.5(4.4)% 1.69(4.15)/3.35(2.78)% 5.76(27.7)/15.4(16.3)% 18.7(87.6)/17.4(21.6)% 12.7(62.9)/18.9(14.1)% 9.84(10.2)/6.9(6.13)% 25.6(25.7)/15.1(15.5)% 9.2(9.4)/7.2(4.8)% 9.13(8.75)/8.4(6.9)%	/	Antioxidant properties of D/No-D fraction obtained after intestinal digestion of grape/wine (compared to the initial samples) (a) ABTS^•+^ ↓ (b) FRAP ↓ (c) DPPH^•^ ↓ COST INFOGEST protocol of digestion	[140]
Seed and skin extracts (raw and purified)	^2^ Proanthocyanidins (proanthocyanidins of various od degree polymerization (mDP))	(a) Pepsin (315 U/mL) (b) T = 37 °C; pH 1.7; τ = 2 h; 100 rpm	/	(a) Pancreatin (4 mg/mL); bile salt (25 mg/mL) (b) T = 37 °C; τ = 2 h (c) Digest (dialysis; membrane 12 kDa)	Proanthocyanidins in seed and skin extracts were mostly degraded (up to 80%) (seed extracts are more resistant on degradation during digestion)	/	ACE inhibitory activity after intestinal and colon digestion: (a) Raw extracts of seeds and skin were retained activity (b) Purified extracts of seeds and skins after the colon phase had no ACE-inhibitory ability	[141]
Grape Seed Pulp Skin (Bordo (B) and Niagara (N))	^1^ Total phenolics	(a) Pepsin (2000 U/mL) (b) T = 37 °C; pH 3.0; τ = 2 h	/	(a) Pancreatin (800 U/mL) bile salt (25 mg/mL) (b) T = 37 °C; pH 7.0; τ = 2 h (c) Digest (centrifugation)	Bioaccessibility (compared to the initial samples) Grape N (n.s.); B ↑ Skin N (n.s.); B ↑ Pulp N ↑; B ↑ Seed N (n.s.); B ↑	/	Antioxidant properties of digests (compared to the initial samples) (a) Grape ↑; skin ↓; pulp ↑; seed (n.s.) (Niagara) (b) Grape ↑; skin ↑; pulp ↑; seed ↑ (Bordo) COST INFOGEST protocol of digestion	[142]
Grape and wine (mix of two variety Chardonay and Viognier)	^1^ Total phenolics ^2^ Total phenolics ^2^ Catechin	(a) Pepsin (450 U/mL) (b) T = 37 °C; pH 2.0; τ = 2 h	Mix Grape/Wine 37/74% / /	(a) Pancreatin (1.2 mg/g(mL)); bile salt (5.6 mg/g(mL)) (b) T = 37 °C; pH 7.5; τ = 2 h (c) Digest (dialysis)	Grape No-D(D)/Wine No-D(D) 13(18)/34(33)% 1.60(3.85)/40.9(30.3)% 0(0)/10.25(0)%	/	Antioxidant properties of D/No-D fractions after intestinal digestion of grape/wine (compared to the initial samples) (a) ABTS^•+^ ↓ (b) FRAP ↓ (c) DPPH^•^ ↓	[143]
Skin extract	^2^ Total phenolics ^2^ Procyanidin B1 ^2^ Catechin ^2^ Quercetin-3-glucoside ^2^ Quercetin-3-glucuronide ^2^ Caftaric acid ^2^ Coutaric acid	(a) Pepsin (40 mg/mL) (b) T = 37 °C; pH 2.5; τ = 1 h	/	(a) “Porcine” lipase (2 mg/mL); pancreatin (4 mg/mL); bile salt (24 mg/mL) (b) T = 37 °C; pH 5.3–6.5; τ = 2 h (c) Digest (centrifugation)	65.59% 47.31% 53.82% 43.31% 51.79% 80.21% 66.77%	/	Skin extracts after in vitro digestion have shown reduced influence on HT-29 intestinal cells in ROS and GSH modulation assessed	[144]
Wine	^2^ Trans-resveratrol ^2^ Chlorogenic acid ^2^ Caffeic acid ^2^ p-coumaric acid ^2^ 3-(4-hydroxyphenyl) propionic acid	(a) Pepsin (40 mg/mL) (b) T = 37 °C; pH 2.0; τ = 2 h	White/Red wine 80.5/120.2% 40/81.8% 122.3/79.1% 87.8/96.8% n.d/n.d	(a) Pancreatin (0.02 g); bile salt (0.12 g) (b) T = 37 °C; pH 6.0–7.4; τ = 2.5 h (c) “colon” phase (presence of microflora)	White/Red wine 74.4/117.9% 23.1/68.9% 133.9/65.6% 82.0/91.8% n.d./100% (13.23 µg/mL)	White/Red wine 67.2/56.86% 12.3/4.19% 11.6/128.1% 78.02/86.3% n.d./100% (9.80 µg/mL)	Antioxidant activity after intestinal digestion and colon fermentation (compared to initial samples) (a) ABTS^•+^ ↓	[145]
Freeze-dried sediment from grape juice (Isabel (I) and Bordo (B))	^2^ Total phenolics ^2^ Total phenolic acid ^2^ Total flavan-3-ols ^2^ Catechin ^2^ Epicatechin ^2^ Total anthocyanins ^2^ Malvidin-3,5-diglucoside	(a) Pepsin (2000 U/mL) (b) T = 37 °C; pH 3,0; τ = 2 h	Bordo/Isabel sediments 95.3/142.6% 132.2/149.9% 98.7/150% 97.2/135.8% 100/196.9% 84.6/131.7% 86.8/111.1%	(a) Pancreatin (800 U/mL); žučne soli (10 mol/L) (b) T = 37 °C; pH 7.0; τ = 2 h (c) Digest (centrifugation)	Bordo/Isabel 54.9/14.4% 71.9/65.5% 49/14.2% 49.8/2.0% 48.1/54.5% 57.5/8.0% 59.8/0.1%	/	Antioxidant properties of D/No-D fractions after intestinal digestion of grape juice sediments (compared to the initial samples) (a) ABTS^•+^ ↓ (b) DPPH^•^ ↓ COST INFOGEST protocol of digestion	[146]
Wine (Cabernet Sauvignon (CS); Chardonnay (C))	^2^ Total phenolics	(a) Pepsin (450 U/mL) (b) T = 37 °C; pH 1.2; τ = 2 h	CS/C wine 118.3/82.8% (low consumption) 124.1/83.9% (moderate consumption) 131.6/89.8% (Excessive consumption)	(a) Pancreatin (4 mg/mL); bile salt (25 mg/mL) (b) T = 37 °C; pH 7.5; τ = 2 h (c) Digest (dialysis; passive diffusion; membrane 12 kDa) (d) “colon” fermentation	CS/C wine 54.0/43.7% (low consumption) 56.2/43.7% (moderate consumption) 74.0/45.7% (Excessive consumption)	CS/C wine 27.2/20.6% (low consumption) 30.4/22.3% (moderate consumption) 41.6/25.0% (excessive consumption)	Functional properties after colon fermentation (compared to the initial samples) (a) α-glucosidase inhibitory activity ↓ (b) α-amylase inhibitory activity ↓ COST INFOGEST protocol of digestion	[147]
Pomace (Po), skin (Sk), peduncle (Pe), and seed (Se) extracts	^1^ Total phenolics ^2^ Total phenolics ^2^ Total phenolic acid ^2^ Total flavanols ^2^ Catechin ^2^ Epicatechin ^2^ Total flavonols ^2^ Quercetin-3-glucoside	(a) “porcine” pepsin (b) T = 37 °C; pH 3.0; τ = 2 h	Po/Sk/Pe/Se 31.3/79.3/93.8/34.6% 128/233/140/123% / 118/0/156/99% 125.8/197.8% (Se/Pe) 102.8% (Se) 30/47/27/0% 40.6% (Pe)	(a) Pancreatin (800 U/mL); bile salt extracts (20 mg) (b) T = 37 °C; pH 7.5; τ = 2 h	Po/Sk/Pe/Se 49.8/101/112.9/38.7% 156.4/58.44/153.5/91.4% 75/355/92/62% 228/0/201/97% 110.3/330.9% (Se/Pe) 101.2% (Se) 47/72/48/0% 70.1% (Pe)	/	Antioxidant properties of Po, Sk, Pe, and Se after intestinal digestion (compared to the initial samples) (a) ORAC↓ (b) DPPH^•^ ↓ COST INFOGEST protocol of digestion	[148]
Cane (Ca) and peduncle (bunch; Pe) (cv Malbec)	^1^ Total phenolics ^2^ Total phenolic acid ^2^ Stilbenes ^2^ Ɛ-viniferin ^2^ Catechin ^2^ Total flavonols	(a) “porcine” pepsin (b) T = 37 °C; pH 2.0; τ = 1 h	Cane/Peduncle 58.9/80.4% 124.3/92.8% 44.3/0% 46.6/0% 73.2/88.1% 51.8/79.4%	(a) Pancreatin; bile salt extracts (20 g/L) (b) T = 37 °C; pH 6.8; τ = 2 h	Cane/Peduncle 73.9/20.3% 87.6/27.2 129.9/51.6% 136.6/51.6% 29.6/12.1% 52.2/61.9%	/	Antioxidant properties of cane and peduncle after intestinal digestion (compared to the initial samples) (a) ORAC: cane ↑ and peduncle (n.s) (b) DPPH^•^: cane ↓ and peduncle ↓	[64]
Pomace (Tempranillo)	^2^ Total phenolics ^2^ Gallic acid ^2^ Ellagic acid ^2^ Total flavonols ^2^ Total anthocyanins	(a) “porcine” pepsin (b) T = 37 °C; pH 2.0; τ = 2 h	96.8% 100.7% 99.3% 88.8% 65.8%	(a) Pancreatin; bile salt extracts (12 mg/mL) (b) T = 37 °C; pH 7.5; τ = 2 h (c)Digests (centrifugation)	85.5% 96.4% 86.2% 66.1% 0%	/	Antioxidant properties after intestinal digestion of pomace (compared to the initial samples) (a) ABTS^•+^ ↓ (b) FRAP ↓ (c) ORAC ↓	[106]
Raisins (Sultana)	^1^ Total phenolics	(a) Pepsin (b) T = 37 °C; pH 1.7; τ = 2 h; 100 rpm	97.88%	(a) Pancreatin (4 mg/mL); bile salt (25 mg/mL) (b) T = 37 °C; pH 7.0; τ = 2 h (c) Digest (dialysis)	7.86% (D fraction) 102.5% (No-D fraction)	/	Antioxidant properties of D/No-D fractions after intestinal digestion of raisins (compared to the initial samples) (a) ABTS^• +^ : D ↓ and No-D↓ (b) DPPH^•^: D ↓ and No-D↑ (c) FRAP: D ↓ and No-D ↑ (d) CUPRAC: D ↓ and No-D↑	[149]
Grape seed	^1^ Total phenolics ^2^ Total phenolics ^2^ Gallic acid ^2^ Catechin-hydrat ^2^ Epicatechin	(a) “porcine” pepsin (b) T = 37 °C; pH 2.0; τ = 1 h; 95 rpm	Red/White variety 140.9/107.4% 25.9/81.8% 65.2/137.4% 29.7/47.2% 31.4/55.8%	(a) Pancreatin (4 mg/mL); glycodeoxycholate (0.04 g/mL); taurodeoxycholate (0.025 g/mL) (b) T = 37 °C; pH 7.4; τ = 2 h	Red/White variety 55.5/107% 33.4/25.5% n.d. 24/0% 11.8/0%	/	Antioxidant properties of seed after intestinal digestion (compared to the initial samples) (a) ABTS^•+^ ↓ (b) FRAP ↓ (c) DPPH^•^ ↓	[150]
Grape pomace extract (Croatina variety)	^1^ Total phenolics ^1^ Total anthocyanins ^2^ Ellagic acid ^2^ Catechin ^2^ Resveratrol	(a) Pepsin solution (0.4 mL; 2000 U/mL) (b) T = 37 °C; pH 3.0; τ = 2 h	99.5% 77.4% 162.6% / 80.2%	(a) Pancreatin solution (trypsin activity 100 U/mL); bile solution (10 mmol/L) (b) T = 37 °C; pH 7.0; τ = 2 h (c) Digest (filtration 0.22 µm)	104.7% 17.1% 209.5% 0% 66.4%	/	Antioxidant properties of grape pomace extract after intestinal digestion (compared to the initial non-digested sample) (b) FRAP ↑ (c) DPPH^•^ ↑ COST INFOGEST protocol of digestion	[151]
Fresh black “Isabel” grape	^1^ Total phenolics ^1^ Total anthocyanins ^2^ Gallic acid ^2^ Catechin ^2^ Malvidin-3-O-glucosdie	(a) Pepsin solution (2000 U/mL) (b) T = 37 °C; pH 3.0; τ = 2 h	28.1% 21.2% 48.8% 57.3% 3.23%	(a) Pancreatin solution (100 U/mL); bile solution (10 mmol/L) (b) T = 37 °C; pH 7.0; τ = 2 h (c) Dialysis and centrifugation	D/No-D 15.1/17.9% 0.67/0.44% 0/35.4% 0.56/0.28% n.d./n.d. (0%)	/	Antioxidant properties of D/No-D fractions after intestinal digestion (compared to the initial samples) (a) ABTS^• +^ : D ↓ and No-D ↓ (b) DPPH^•^: D ↓ and No-D ↓ (c) FRAP: D ↓ and No-D ↓ (d) CUPRAC: D ↓ and No-D ↓ COST INFOGEST protocol of digestion	[152]
Grape juice and wine (Cabernet Sauvignon)	^1^ Total phenolics ^1^ Total anthocyanins ^2^ Catechin ^2^ Epicatechin ^2^ Resveratrol ^2^ Malvidin-3-O-glucosdie	(a) Pepsin (0.025 g of pepsin) (b) T = 37 °C; pH 3.0; τ = 1 h	Grape/Wine 100.9/92.6% 82.1%/90.8% 87.4/95.3% 58.4/89.9% 0/109.3% 94.9/95.4%	(a) Pancreatin (0.025 g of pancreatin); bile solution (10 mmol/L) (b) T = 37 °C; pH 7.0; τ = 2 h; moderate shaking	Grape/Wine 90.8/87.9% 34.1/60.0% 74.7/83.2% 44.9/64.6% 0/78.1% 4.73/13.0%		Antioxidant, acetylcholinesterase (AChE) and angiotensin-I converting enzyme (ACE) inhibition potential of grape juice and wine after intestinal digestion (compared to the initial non-digested samples) a) DPPH^•^: grape juice ↓ and wine ↓ (b) Inhibition of LP (lipoprotein): grape juice (n.s.) and wine ↑ (c) HO^•^: grape juice ↑ and wine ↑ (d) AChE: grape juice ↓ and wine ↓ (e) ACE: grape juice ↑ and wine ↑ COST INFOGEST protocol of digestion	[153]
Skin extract (Alicante Bouschet variety)	^1^ Total phenolics ^1^ Total anthocyanins ^2^ Catechin ^2^ Epicatechin ^2^ Malvidin-3-O-glucosdie	(a) Pepsin solution (40 mg/mL) (b) T = 37 °C; pH 2.5; τ = 1 h, 60 rpm	49.8% 125.3% 86.3% 90.4% 48.04%	(a) Pancreatin–lipase solution (10 mg/mL of pancreatin and 5 mg/mL of lipase); bile solution (40 mg/mL) (b) T = 37 °C; pH 6.5; τ = 2 h; 60 rpm (b) Colonic fermentation.	48.03% 28.8% 26.3% 23.1% 30.3%	0.33% n.d. n.d. n.d. n.d.	Antioxidant properties of skin extract after intestinal digestion and colonic fermentation (compared to the initial non-digested sample) (a) ABTS^•+^ ↑ (b) ORAC ↑	[154]
Skin (Sk) and seed (Se) extracts (Prokupac variety)	^2^ Ellagic acid ^2^ Total monomeric flavan-3-ols ^2^ Total procyanidin A type ^2^ Total procyanidin B type ^2^ Total procyanidin gallate ^2^ Total flavan-3ols and procyanidins ^2^ Malvidin-3-O-glucoside ^2^ Peonidin-3-O-glucoside	(a) Pepsin solution (1.6 mL, 25,000 U/mL) (b) T = 37 °C; pH 3.0; τ = 2 h, 300 rpm	/	(a) Pancreatin solution (5 Ml, 800 U/mL trypsin activity); 160 mM bile solution (2.5 mL) (b) T = 37 °C; pH 7.0; τ = 2 h; 300 rpm (b) Digestion of supernatans digests (centrifugation; filtration)	Skin/Seed 83.0/41.6% –/0.3% –/3.0% –/0.1% –/0.5% –/0.2% 22.3/– % 0/– %	/	Antioxidant properties of skin and seed extract after intestinal digestion (compared to the initial non-digested sample) (a) ABTS^•+^ ↑ (b) FRP ↓ (a) FCC ↑ (due to presence of enzymes) COST INFOGEST protocol of digestion	[33]

^1^ Spectrophotometric analysis; ^2^ HPLC quantification; n.d.—not detected; n.s.—no significant difference. Non-dialyzable (No-D) fraction after intestinal digestion (potentially colon-available fraction); dialyzable (D) fraction after intestinal digestion (fraction passing through the dialysis membrane). Antioxidant assays: ABTS^•+^ radical scavenging activity (ABTS^• +^); DPPH^•^ radical scavenging activity (DPPH^•^); FRAP—ferric reducing antioxidant power; ORAC—oxygen radical absorbance capacity; CUPRAC—cupric ion reducing antioxidant capacity; TEAC—Trolox equivalent antioxidant capacity; HO^•^—scavenging of hydroxyl radicals; AChE—acetylcholinesterase inhibition potential; ACE—angiotensin-I converting enzyme inhibition potential (cardioprotective properties); FRP—ferric ion reducing power; FCC—ferrous ion chelating capacity; “↑”—increased bioaccessibility of phenolic compounds; : “↓”—decreased bioaccessibility of phenolic compounds

## 7. Bioaccessibility of Grape-Derived Phenolic Compounds

The bioaccessibility of total and individual grape-derived phenolic compounds after in vitro gastric, intestinal, and colonic digestion is summarized in Table 1. The in vitro bioaccessibility of phenolic compounds from solid grape matrices depends primarily on the berry constituents and the release rate of phenolic compounds during digestion. The presence and content of phenolic compounds vary from grape variety to grape variety, which may be reflected in their bioaccessibility during digestion. Red grape varieties generally have a higher content of total and individual phenolic compounds and various anthocyanin derivatives than white grape varieties. However, Gomes et al. [142] showed an increased bioaccessibility of total phenolics after gastrointestinal digestion of “Niagara” (white) grapes, skins, and seeds, whereas bioaccessibility after gastrointestinal digestion was not statistically significant for the red “Bordo” variety. These bioaccessibility differences can be related to the structural variability of these grape varieties. Only phenolic compounds that are released in the intestinal tract can potentially be bioaccessible in the small intestine. Grape berries and pomace contain dietary fibers, proteins, and lipids that can bind phenolic compounds and modulate their bioaccessibility [98]. Dietary fibers contained in the structure of grape skin and seeds effectively interact with the hydroxyl groups of phenolic compounds, reducing their release in the gastrointestinal cocktail, their bioaccessibility, and their antioxidant activity [23]. During the oral in vitro phase, often, only a small portion of the phenolic compounds is released, which remains in the structure of the berry, skin, or seed. Phenolic compounds from solid matrices (berries or pomace) are released more intensively during in vitro gastric and intestinal digestion due to different pH conditions or even in the colon phase due to the action of microorganisms that degrade the plant matrix [115]. On the other hand, the bioaccessibility of phenolic compounds during the digestion of wine or grape/pomace extracts directly depends on the pH conditions in the gastrointestinal phase, as well as on the ability of phenolic compounds to interact with enzymes and compounds of the digestive cocktail. The presence, variability, and content of phenolic compounds in wine/pomace extracts also depend on the grape varieties and the winemaking process (distribution of phenolics from the grape to the wine). Sun et al. [147] showed that wines from different grape varieties have different degrees of bioaccessibility with respect to total and individual phenolic compounds, while Lingua et al. [143] showed different degrees of bioaccessibility for phenolics from grapes (a blend of “Chardonay” and “Viognier”) and their wine. Phenolic compounds from grape pomace extracts have been shown to have better gastric stability and higher bioaccessibility after the intestinal phase compared to phenolic compounds from pomace flour [136]. Several studies have shown that the total content of total phenolic acids, flavonols, flavan-3-ols, or selected phenolic compounds (catechin, epicatechin, and procyanidins) increases after gastric digestion of grapes and pomace compared to undigested samples [113,140,146,148,150]. This increase in the content of phenolic compounds in the gastric phase is attributed to an increased release of phenolic compounds from the matrix, which is probably caused by the “acidic” environment and the action of the digestive enzymes [150]. At the end of the intestinal phase, the bioaccessibility of most phenolic compounds has been reduced. Numerous studies have shown a reduced bioaccessibility of flavonols (quercetin, kaempferol, isorhamnetin, myricetin, syringetin, laricitrin, and their glycosides) and phenolic acids [106,140,144,148]. Anthocyanins are mostly stable in the gastric phase as they are present in the form of red flavylium ions under low-pH conditions [146]. In the intestinal phase, however, the anthocyanins transform into colorless carbinol forms due to the altered pH conditions, which limits their detection and the assessment of bioaccessibility [114]. Low bioaccessibility of total anthocyanins was confirmed after the intestinal digestion of grapes [115,140], wine [135], pomace [106,139], and freeze-dried grape juice sediment [146]. Low bioaccessibility was also confirmed for individual anthocyanins after intestinal digestion. For example, Lingua and Wunderlin [140] showed that the bioaccessibility of peonidin-3-*O*-glucoside, malvidin-3-*O*-glucoside, and malvidin-3-*O*-acetylglucoside was only 25.6%, 9.4%, and 8.7%, respectively. The lower bioaccessibility of grape catechin and epicatechin after in vitro intestinal digestion can be explained by the lower stability of these compounds under alkaline conditions, as well as their increased tendency to oxidize and polymerize or their ability to interact with digestive enzymes [37,113,143,144,150]. Further, procyanidins have mainly been shown to be less bioaccessible due to their large molecular weight, their ability to interact with enzymes, and their lower solubility in digestive fluids [33,150] (Table 2). In contrast to these observations, José Jara-Palacios and Gonçalves [148] reported an increased bioaccessibility of catechin, epicatechin, procyanidin B1, procyanidin B2, and their gallate after the intestinal digestion of grape pomace. This increase in the bioaccessibility of some monomeric and dimeric flavan-3-ols was probably the result of the degradation of oligomeric (trimeric and tetrameric) procyanidins, which could no longer be detected at the end of digestion. The bioaccessibility of resveratrol varies and depends on the matrix of the grapevine. For example, the bioaccessibility of resveratrol after in vitro gastrointestinal digestion of red/white wine ranged from 56% to 78% [145,153], whereas it was about 66% in grape pomace extract [151]. In addition, Ferreyra and Torres-Palazzolo [64] confirmed an increased *ε*-viniferin content after the digestion of grape skins, probably as a consequence of the depolymerization of oligomeric stilbenoids. Finally, the behavior of grape-derived phenolic compounds under simulated in vitro gastrointestinal conditions is still not fully understood and requires further research using a unified in vitro digestion model.

**Table 2 foods-14-00607-t002:** The effect of different matrices on the bioaccessibility of total or selected grape-derived phenolic compounds from different sources (grapes, grape products, and by-products) and antioxidant properties after in vitro gastrointestinal digestion.

Flour/Extract	Food Matrices	Phenolic Compounds (Effect of Food Matrices on Bioaccessibility of Phenolic Compounds After Intestinal Digestion)	Antioxidant Properties After Intestinal Digestion	Ref.
Grape extract (Eminol^®^)	Control (extract + water) (a) Milkshake (MS) (b) Custard dessert (CD) (c) Pancake (PA) (d) Omelette (OM)	Total bioaccessibility (bioaccessible phenolic compounds in digestible and non-digestible fractions) ^2^ Total anthocyanins (MS ↑; CD n.u.; PA ↑; OM ↑) ^2^ Total proanthocyanidins (MS ↑; CD ↑; PA ↓; OM ↑) ^1^ Total phenolics (MS ↓; CD ↓; PA ↓; OM n.b.)	(Observed in comparison to the same sample after the oral phase) (a) FRAP—Ferric reducing antioxidant power Digestible fraction (MS ↓; CD ↓; PA ↑; OM ↑) Non-digestible fraction (MS ↑; CD ↑; PA ↓; OM ↑) Sum of fractions (MS n.s.; CD n.s.; PA ↑; OM ↑) (b) ORAC—Oxygen radical absorbance capacity Digestible fraction (MS ↑; CD ↑; PA ↑; OM ↑) Non-digestible fraction (MS ↑; CD ↑; PA ↑; OM ↑) Sum of fractions (MŠ ↑; KD ↑; P ↑; O ↑)	[35]
Grape extract (Eminol^®^)	Control (extract + water) (a) Milkshake (MS) (b) Custard dessert (CD) (c) Pancake (PA) (d) Omelette (OM)	Total bioaccessibility (bioaccessible phenolic compounds in digestible and non-digestible fractions) ^2^ Delphinidin-3-O-glucoside (MS ↑; CD n.b.; PA ↑; OM ↑) ^2^ Cyanidin-3-O-glucoside (MS ↑; CD n.b.; PA ↑; OM ↑) ^2^ Petunidin-3-O-glucoside (MS ↑; CD ↑; PA ↑; OM ↑) ^2^ Peonidin-3-O-glucoside (MS ↑; CD ↑; PA ↑; OM ↑) ^2^ Malvidin-3-O-glucoside (MS ↑; CD ↑; PA ↑; OM ↑) ^2^ Metylpiranomalvidin-3-O-glucoside (MS ↑; CD ↑; PA ↑; OM ↑) ^2^ Peonidin-3-O-acetylglucoside (MS ↑; CD ↑; PA ↑; OM ↑) ^2^ Delphinidin-3-O-coumaroylglucoside (n.d.) ^2^ Malvidin-3-O-acetylglucoside (MS ↑; CD ↑; PA ↑; OM ↑) ^2^ Petunidin-3-O-coumaroylglucoside (MS n.b.; CD n.b.; PA ↑; OM ↑) ^2^ Peonidin-3-O-coumaroylglucoside (MS n.b.; CD n.b.; PA ↑; OM ↑) ^2^ Malvidin-3-O-coumaroylglucoside (MS n.b.; CD n.b.; PA ↑; OM ↑)	/	[34]
Grape extract (Eminol^®^)	(a) Biscuits (b) Buns, breadsticks	Bioaccessibility (calculated in comparison to the same sample before digestion) (a) Biscuits + anthocyanins ↓ (57.26%) (b) Biscuits + anthocyanins + docosahexaenoic acid ↓ (8.83%) (c) Buns + anthocyanins ↓ (57.30%) (d) Buns + anthocyanins + docosahexaenoic acid ↓ (n.d.)	/	[36]
Grape pomace powder	Bread with 5.0% (GP5) and 10.0% (GP10) grape pomace powder	Bioaccessibility (calculated in comparison to the same sample before digestion) ^2^ Anthocyanins ↓ (GP5, 5.88%; GP10, 7.25%) ^2^ Flavones ↓ (GP5, 9.33%; GP10, 6.74%) ^2^ Phenolic acid ↓ (GP5, 0.94%; GP10, 1.25%)	Antioxidant properties of bread with grape pomace powder after intestinal digestion (compared to the initial non-digested the same bread with grape pomace powder) (a) ABTS^•+^ radical scavenging activity ↓ (for both GP5 and GP10) (b) FRAP—Ferric ion reducing power ↓ (for both GP5 and GP10)	[155]
Skin (Sk) and seed (Se) extracts (Prokupac variety)	Infant puree (Juvitana, Swisslion Product d.o.o. Indjija, Serbia) (boiled turkey meat (20%); boiled potato paste (10%; boiled corn paste (25%); rice flour (5%); 0.1% NaCl and water (39.9%))	Bioaccessibility (digested skin/seed extract with food matrix-total recovery) (calculated in comparison to the non-digested initial skin and seed extract) ^2^ Ellagic acid ↓ (Skin, 75.4% and Seed, 57.1%) ^2^ Total monomeric flavan-3-ols ↓ (Seed, 5.7%) ^2^ Total procyanidin A type ↓ (Seed, 4.5%) ^2^ Total procyanidin B type ↓ (Seed, 2.7%) ^2^ Total procyanidin gallate ↓ (Seed, 8.1%) ^2^ Total flavan-3ols and procyanidins ↓ (Seed, 4.7%) ^2^ Malvidin-3-O-glucoside ↓ (Skin, 0%) ^2^ Peonidin-3-O-glucoside ↓ (Skin, 0%)	Antioxidant properties of digested skin/seed extract with food matrix (compared to the initial non-digested skin/seed extracts) (a) ABTS^•+^ radical scavenging activity ↑ (for both seed and skin) (b) FRP—Ferric ion reducing power ↓ (for both seed and skin) (c) FCC—Ferrous ion chelating properties ↑ (for both seed and skin) (contributed by enzymes and food matrices)	[33]
Seed extract (Prokupac variety)	Thermally treated goat′s milk powder (goat milk fortified with 0.6 mg TPC per mL milk)	Bioaccessibility (digested seed/milk powder-total recovery) (calculated in comparison to the non-digested initial seed extract) ^2^ Gallic acid ↓ (0%) ^2^ Catechin ↓ (26.19%) ^2^ Catechin gallate ↓ (4.66%)	Antioxidant properties of digested seed/milk powder (compared to the initial non-digested seed/milk powder) (a) ABTS^•+^ radical scavenging activity ↑ (b) FRP—Ferric ion reducing power ↓ (c) FCC—Ferrous ion chelating properties ↑ (d) TAC—Total antioxidant capacity (in vitro phosphomolybdenum reducing capacity) ↓	[37]
Red wine (RW) and white wine (WW)	Juvitana infant formula (20% turkey meat; 25% corn paste; 10% potato paste; 5% rice flour; 0.1% NaCl; water)	^1^ Total phenolics LC (RW n.b.; WW n.b.); MC (RW n.b.; WW n.b.); HC (RW ↓; WW n.b.) ^1^ Total anthocyanins LC (RW n.b.); MC (RW n.b.); HC (RW ↑) ^2^ Total flavan-3-ols LC (RW ↑; WW n.b.); MC (RW n.b.; WW n.b.); HC (RW n.b.; WW n.b.) ^2^ Total phenolic acid LC (RW n.b.; WW n.b.); MC (RW ↓; WW n.b.); HC (RW ↑; WW n.b.)	(observed in comparison to the wine sample without matrix after intestinal digestion) (a) DPPH^•^ radical scavenging activity LC (RW n.s.; WW ↑); MC (RW n.s.; WW n.s.); HC (RW ↑; WW n.s.) (b) ABTS^•+^ radical scavenging activity LC (RW ↓; WW n.s.); MC (RW n.s.; WW ↑); HC (RW ↑; WW n.s.) (c) ORAC—Oxygen radical absorbance capacity LC (RW ↑; WW ↑); MC (RW ↑; WW ↑); HC (RW ↑; WW n.s.) (d) FRAP—Ferric reducing antioxidant power LC (RW n.s.; WW n.s.); MC (RW ↑; WW n.s.); HC (RW ↑; WW n.s.)	[147]
(a) Isabel grape (IG) (b) Isabel grape + pomace flour (IGF)	Goat′s milk (probiotic yoghurt)	/	(observed in comparison to the same sample before digestion) Model digestions with enzymes (a) ABTS^•+^ radical scavenging activity Yoghurt + IG ↑; yoghurt + IGF ↑ (b) ORAC—Oxygen radical absorbance capacity Yoghurt + IG ↑; yoghurt + IGF ↑	[156]
Grape juice (GJ)	Cow′s milk (a) Whole milk (WM) (b) Skimmed milk (SM)	^1^ Total phenolics (WMGJ ↑; SMGJ ↑) ^2^ Caffeoyltartaric acid (WMGJ ↓; SMGJ ↓) ^2^ Epicatechin (WMGJ ↓; SMGJ ↓) ^2^ Proantocyanidins (WMGJ ↑; SMGJ ↓) ^2^ Protocatechuic acid glucoside (WMGJ ↑; SMGJ ↑)	(observed in comparison to the same sample before digestion) (a) ABTS^•+^ radical scavenging activity Whole milk + grape juice ↑ (WMGJ) Skimmed milk + grape juice ↑ (SMGJ)	[157]
Grape juice (Concord variety)	(a) Ultrafiltrated permeate (b) Acid whey (c) Whole milk (d) Skimmed milk	/	(observed in comparison to the same sample before digestion) (a) FRAP—Ferric reducing antioxidant power Ultrafiltrated permeate + grape juice (n.s.) Acid whey + grape juice (n.s.) Whole milk + grape juice ↑ Skimmed milk + grape juice (n.s.)	[158]
Fruit mix (with a portion of Airen grape concentrate) (FM)	Skimmed milk	^1^ Total phenolics Fruit mix + milk ↑ Fruit mix + Fe + milk ↑ Fruit milk + Zn + milk (n.b.) Fruit mix + Fe + Zn + milk (n.b.)	(observed in comparison to the same sample before digestion) Fruit mix + milk (ORAC↑; ABTS^•+^ ↑) Fruit mix + Fe + milk (ORAC↑; ABTS^•+^ ↑) Fruit mix + Zn + milk (ORAC↑; ABTS^•+^ ↑) Fruit mix + Fe + Zn + milk (ORAC↑; ABTS^•+^ ↑)	[138]
Fruit beverages (with a portion of grape concentrate) (FB)	Skimmed milk	Fruit beverages + Fe + milk ^2^ Total phenolics ↓ ^2^ Hydroxycinnamic acid derivatives ↓ ^2^ Total flavons ↓ ^2^ Total flavan-3-ols ↓ ^2^ Total flavanons ↓	/	[159]

^1^ Spectrophotometric analysis; ^2^ HPLC quantification. n.s.—no significant difference among results for applied antioxidant test; n.d.—not detected; n.b.—no effect on bioaccessibility of grape phenolic compounds; “↑”—increased bioaccessibility of phenolic compounds/higher antioxidant activity; “↓”—decreased bioaccessibility of phenolic compounds/lower antioxidant activity; LC—low wine consumption; MC—moderate wine consumption; HC—high wine consumption.

### Effect of Food Matrices on Bioaccessibility of Grape-Derived Phenolic Compounds

Investigations on in vitro gastrointestinal digestion and bioaccessibility assessment of phenolic compounds were mainly performed in naturally phenolics-enriched food matrices such as fruits, vegetables [160], or grape-based products [106,144,148]. Only a few studies have investigated the influence of food matrices on the bioaccessibility of phenolic compounds (Table 2). Sengul and Surek [161] published an interesting model study showing how matrices of different foods and individual food components influence the bioaccessibility of phenolic compounds, especially anthocyanins. The analysis of the individual food constituents showed that proteins (soy protein, casein, and meat proteins) contribute to the reduction in the total amount of anthocyanins. On the other hand, some carbohydrates and fatty acids (stearic and linolenic acids) showed a protective effect on anthocyanins and their content in the soluble fraction during/after in vitro gastrointestinal digestion. Dietary fibers also have an effect on the bioaccessibility of phenolic compounds during in vitro digestion. Phenolic compounds most commonly bind to the surface of the fibers and are partially released in the gastric and intestinal phases [37]. Unabsorbed phenolic compounds that remain on the surface of the fibers are potentially available in colonic phase and are released by the action of bacteria [23]. In recent years, the bioaccessibility of grape-derived phenolic compounds in the presence of complex (not naturally rich in phenolic compounds) food matrices, such as dairy, egg, and bakery products, as well as oatmeal formulations and infant formulas, has been increasingly studied [33,34,35,36,37,161,162,163]. Sengul and Surek [161] have shown that complex food matrices (milk, bread, and yoghurt) affect the reduction in total phenolic compounds and anthocyanins after in vitro gastrointestinal digestion in dialyzable and non-dialyzable fractions. Pineda-Vadillo and Nau [35] have shown that the addition of grape extract to milk- and egg-based food matrices has significant effects on the release and solubility of anthocyanins and procyanidins during digestion, especially in solid matrices (pancakes and omelettes). The same authors have suggested that the incorporation of grape anthocyanins into food matrices is an effective way to protect them from gastrointestinal degradation and a promising model for targeted delivery [34]. The effect of complex food matrices containing meat and carbohydrates on the bioaccessibility of phenolic compounds during in vitro gastrointestinal digestion has thus far been poorly analyzed [33,147,163,164]. Pešić abd Milinčić [33] reported on the low recovery rate of total and individual proanthocyanidins and anthocyanins in the presence of cereal- and meat-based matrices (i.e., infant formula simulating a standard meal) after in vitro gastrointestinal digestion and suggested a partial protective effect of this food matrix on proanthocyanidins. On the other hand, using the same food matrix, Sun and Cheng [147] demonstrated the high bioaccessibility of phenolic compounds from wine after the colon phase of gastrointestinal digestion and strong biological activity. The above-mentioned studies suggest that food matrices enriched with grape-derived phenolic compounds have increased antioxidant activity. This is due to the joint contribution of the released phenolic compounds and the hydrolysates from the food matrices, presenting a good strategy for formulating a healthier diet [33,35,37,164]. Bakery products enriched with grape skin extract have shown high bioaccessibility of anthocyanins at the end of gastrointestinal digestion. However, the bioaccessibility of anthocyanins was significantly reduced by the addition of docosahexaenoic acid to the biscuit and buns formulations [36]. On the other hand, Rocchetti and Rizzi [155] reported a very low recovery of phenolic compounds (anthocyanins, flavones, and phenolic acids) after the in vitro gastrointestinal digestion of bread containing grape pomace flour. Due to the contradictory results and the limited number of studies addressing this issue, further research is needed to evaluate the effects of complex food matrices on the bioaccessibility of grape-derived phenolic compounds.

## 8. Conclusions

Grape-derived phenolic compounds have been studied extensively, mainly because of their positive effects on human health. However, phenolic compounds undergo transformations during digestion, limiting their bioaccessibility and altering their biological activity. The bioaccessibility of grape-derived phenolic compounds is a crucial step in their absorption and activity. Phenolic compounds are very sensitive to slight changes in digestive parameters, which is reflected in their bioaccessibility. In this context, this review considered digestion models and factors affecting the bioaccessibility of grape-derived phenolic compounds with the aim of understanding the behavior of phenolic compounds during digestion and promoting the development of new food products with targeted bioaccessibility. So far, different types of in vitro static gastrointestinal digestion have mainly been used to evaluate the bioaccessibility of grape phenolic compounds, focusing on individual anthocyanins, flavan-3-ols, procyanidins, and phenolic acid esters. Moreover, the bioaccessibility of these compounds is often variable and highly dependent on their presence in the grape, skin, seed, or wine. Applied digestion models have shown that the physiological conditions of the different digestion phases, the physicochemical properties of the phenolic molecules, and the presence of food matrices significantly influence the bioaccessibility of grape-derived phenolic compounds. Due to the different polarity, the balance between hydrophilicity and lipophilicity is crucial for the solubilization/micelarization and recovery of grape-derived phenolic compounds in the intestinal phase. In addition, the interactions of phenolic compounds with food macroconstituents (proteins, lipids, and carbohydrates) and enzymes alter the digestibility of food, as well as the bioaccessibility and antioxidant activity of phenolic compounds. Anthocyanins are pH-sensitive and usually have low intestinal bioaccessibility, as do procyanidins, which have a high tendency to react with enzymes and food constituents. However, the effects of complex food matrices on the bioaccessibility of grape-derived phenolic compounds are still insufficiently investigated, and further studies are needed. In the colon phase, microorganisms convert phenolic compounds into metabolites that are further absorbed and metabolized. However, studies involving the colonic phase are rare, and analyzing phenolic metabolites in vivo is a major challenge for researchers. This review summarizes previous studies on this topic and aims to help standardize the digestion protocol for evaluating the bioaccessibility of grape-derived phenolic compounds, which would allow for better comparability of the results from different studies. In addition, this review should promote the development of novel food products with targeted bioaccessibility of grape-derived phenolic compounds.

## Figures and Tables

**Figure 2 foods-14-00607-f002:**
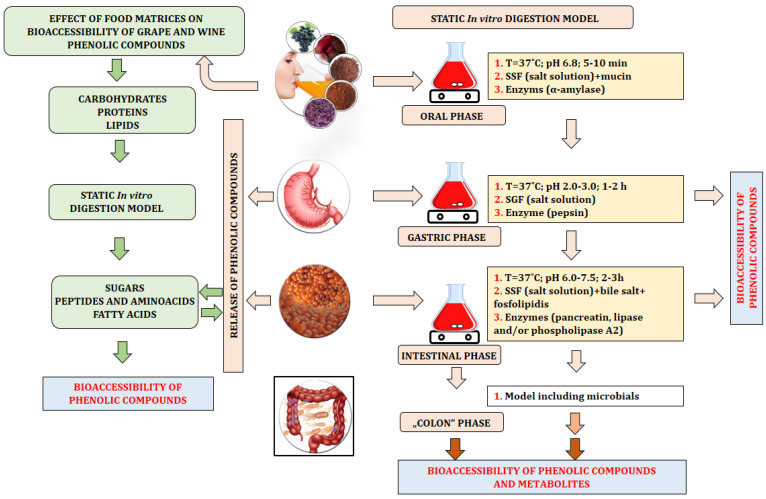
Schematic representation of static in vitro digestion model for the evaluation of bioaccessibility of grape-derived phenolic compounds with/without of food matrix.

## Data Availability

No new data were created or analyzed in this study. Data sharing is not applicable to this article.

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
