# Peer review of "The Bioaccessibility of Grape-Derived Phenolic Compounds: An Overview"

_foods, 2025, doi:10.3390/foods14040607_

Round 1

Reviewer 1 Report

Comments and Suggestions for Authors

It is important to include a section that indicated the methodology that was followed for the search and selection of the articles that were used to prepare the review.

Make a figure or table indicating the main secondary metabolites (bioactive compounds) present in grapes, wine and by-products. This helps to identify them and facilitate reading the manuscript.

Author Response

Thank you very much for recognising the importance and scientific value of our review. Thank you also for your valuable suggestions and the opportunity to respond to your comments. All corrections are marked in red in the revised manuscript.

  1. It is important to include a section that indicated the methodology that was followed for the search and selection of the articles that were used to prepare the review.

Thank you for the suggestion. The methodology for the literature review is included in the manuscript. Section 2 in the revised manuscript.

  1. Make a figure or table indicating the main secondary metabolites (bioactive compounds) present in grapes, wine and by-products. This helps to identify them and facilitate reading the manuscript.

Thank you for the suggestion. The figure showing the main phenolic compounds of grapes, wine and by-products has been added in the revised manuscript. See Figure 1 in the revised manuscript.

Reviewer 2 Report

Comments and Suggestions for Authors

The publication is well-written and reviews research on the bioavailability of polyphenolic compounds from grapes (and products produced from grapes such as wine) during digestion in the gastrointestinal tract.

The publication is divided into subsections addressing the composition of polyphenolic compounds, analysis of factors affecting their bioavailability, analysis of models used to study the digestion of phenolic compounds in the gastrointestinal tract, and the determination of the bioavailability of these compounds, considering the influence of the matrix. According to the intentions, the longest chapters address bioavailability and the models used to analyze the digestion of compounds in the gastrointestinal system. The publication is based on 136 literature sources, thoroughly examining the topic. I recommend the publication for acceptance in the journal Foods. I would only suggest paying attention to the fact that in some places, words such as in vitro or Vitis vinifera are not italicized.

Author Response

The publication is well-written and reviews research on the bioavailability of polyphenolic compounds from grapes (and products produced from grapes such as wine) during digestion in the gastrointestinal tract.

The publication is divided into subsections addressing the composition of polyphenolic compounds, analysis of factors affecting their bioavailability, analysis of models used to study the digestion of phenolic compounds in the gastrointestinal tract, and the determination of the bioavailability of these compounds, considering the influence of the matrix. According to the intentions, the longest chapters address bioavailability and the models used to analyze the digestion of compounds in the gastrointestinal system. The publication is based on 136 literature sources, thoroughly examining the topic. I recommend the publication for acceptance in the journal Foods. I would only suggest paying attention to the fact that in some places, words such as in vitro or Vitis vinifera are not italicized.

Thank you very much for recognising the importance and scientific value of our review. Thank you also for your suggestion. The manuscript has been thoroughly reviewed for typographical errors and corrected accordingly.

Reviewer 3 Report

Comments and Suggestions for Authors

I would like to express my sincere congratulations on the great review.

How does the bioaccessibility of grape-derived phenolic compounds directly influence their bioavailability and biological efficacy in the human body?

What are the main challenges in extracting and stabilizing grape phenolic compounds for application in functional food formulations? Since there are numerous studies on extraction methods (ultrasound, maceration, microwave, ATPS, supercritical, among others) and different solvents (conventional, ionic liquids, eutectic solvents, CO2, among others).

In relation to the chemical variability between different grape cultivars, how do these structural differences impact on the bioaccessibility of phenolic compounds?

How can in vitro digestion models, such as the INFOGEST protocol, be experimentally validated to accurately represent the bioavailability of grape phenolic compounds in humans?

What relationship between grape phenolic compounds and dietary proteins can reduce or increase their bioavailability?

The study mentions the influence of pH on the stability of phenolic compounds. How do variations in the pH of the stomach and intestine modulate the chemical structure and functionality of these compounds?

Author Response

I would like to express my sincere congratulations on the great review.

Thank you very much for acknowledging the importance and scientific value of our review. Thank you also for all suggestions and good will to improve the quality of the manuscript. All corrections are marked in red in the revised manuscript.

  1. How does the bioaccessibility of grape-derived phenolic compounds directly influence their bioavailability and biological efficacy in the human body?

Thank you very much for this question. Your comment has been take into account and included in the revised manuscript. See red marked text in section 5 (4. Factors influencing the bioaccessibility/bioavailability of phenolic compounds).

  1. What are the main challenges in extracting and stabilizing grape phenolic compounds for application in functional food formulations? Since there are numerous studies on extraction methods (ultrasound, maceration, microwave, ATPS, supercritical, among others) and different solvents (conventional, ionic liquids, eutectic solvents, CO2, among others).

In agreement with your comment, a new section (4. Extraction and stabilization of grape-derived phenolic compounds) was added in revised manuscript. See section 4. in revised manuscript.

  1. In relation to the chemical variability between different grape cultivars, how do these structural differences impact on the bioaccessibility of phenolic compounds?

Thank you for your comment. The impact of structural differences between grape cultivars on bioaccessibility of grape/wine phenolic compounds has been considered and added/highlighted in the revised manuscript. See text in red in section 7 (7. Bioaccessibility of grape-derived phenolic compounds).

  1. How can in vitro digestion models, such as the INFOGEST protocol, be experimentally validated to accurately represent the bioavailability of grape phenolic compounds in humans?

Thank you for this valuable comment. A detailed explanation for this commentary has been included in the revised manuscript. Previous studies have shown a good approximation of the harmonized in vitro INFOGEST protocol and the in vivo pig trial, focusing on proteins digestibility. However, assessment of bioaccessibility of various phytochemicals, especially phenolic compounds, is a challenge for researchers, as a slight change in digestion parameters has a significant impact on the results obtained and the results very from laboratory to laboratory. See the text in red in section 6 (6. Models for investigation of gastrointestinal digestion of grape-derived phenolic compounds).

  1. What relationship between grape phenolic compounds and dietary proteins can reduce or increase their bioavailability?

Thank you for this question. Your comment has been taking into account and the explanation has been included in subsection 4.1. of the revised manuscript. See text in red in subsection 5.1. (5.1. The influence of food matrices).

  1. The study mentions the influence of pH on the stability of phenolic compounds. How do variations in the pH of the stomach and intestine modulate the chemical structure and functionality of these compounds?

Thank you for this suggestion. The influence of gastric and intestinal pH on the chemical structure and functionality of phenolic compounds was additionally explained in the revised manuscript. See red marked text in subsection 5.3. (5.3. The influence of physiological conditions encountered in different phases of digestion).

Reviewer 4 Report

Comments and Suggestions for Authors

The manuscript presets a comprehensive review of the bioaccessibility of grape-derived phenolic compounds, focusing on their behavior during gastrointestinal digestion and the factors affecting their bioaccessibility. The review is mainly well structured and clearly expains each topic.

There are however some sugestions to authors:

Are there any other reviews published with the same topic? The authors should briefly present the other reviews and to explain what is the originality of this review.

Line 132: Anthocyanidins are not colorless compounds. Please reconsider this statement or better explain this topic.

The latin name of the species should be written in italic.

There is nothing written in the chemical composition chapter about resveratrol, a polyphenolic compound found in grapes, but it is however mentioned in Line 199.

Line 192-194: I suggest that authors should reconsider the following text: ,, But, the bioaccessibility of phenolic compounds, for example anthocyanins and/or procyanidins is often variable as they tend to readily polymerize or react with enzymes, and constituents of the digestive cocktail’’ since anthocyanins do not undergo polymerization, but rather form complexes with proteins.

The conclusions are overly general and should provide more specific details about the key points discussed in each section of the review.

Comments on the Quality of English Language

The article contains numerous typographical errors; the authors should thoroughly review the entire manuscript and make the necessary corrections.

Author Response

The manuscript presents a comprehensive review of the bioaccessibility of grape-derived phenolic compounds, focusing on their behaviour during gastrointestinal digestion and the factors affecting their bioaccessibility. The review is mainly well structured and clearly explains each topic.

Thank you very much for acknowledging the importance and scientific value of our review. Also, thank you for your valuable suggestions and the opportunity to respond to your comments. All corrections are marked in red in the revised manuscript.

  1. Are there any other reviews published with the same topic? The authors should briefly present the other reviews and to explain what is the originality of this review.

Thank you for this comment. To our knowledge, this is the first review on this topic dealing with the bioaccessibility of grape-derived phenolic compounds. Previous reviews about grape phenolic compounds focus on their health benefits or their extraction from pomace, their valorisation, and/or their application in the food industry. Additional explanations on the originality of this review have been included in the revised manuscript. See the text in red in section 1 (1. Introduction).

  1. Line 132: Anthocyanidins are not colorless compounds. Please reconsider this statement or better explain this topic.

Thank you for this suggestion. We agree with you. This oversight has been corrected in the revised manuscript. See line 164.

  1. The latin name of the species should be written in italic.

The entire revised manuscript has been checked and corrected, in accoradance with your suggestion.

  1. There is nothing written in the chemical composition chapter about resveratrol, a polyphenolic compound found in grapes, but it is however mentioned in Line 199.

Thank you for this suggestion. A new subsection on stilbens and resveratrol has been added in the revised manuscript, providing some additional information on resveratrol. See new subsection 3.2. Stilbens, and the text in red in Section 7, lines 516-519

  1. Line 192-194: I suggest that authors should reconsider the following text: ,, But, the bioaccessibility of phenolic compounds, for example anthocyanins and/or procyanidins is often variable as they tend to readily polymerize or react with enzymes, and constituents of the digestive cocktail’’ since anthocyanins do not undergo polymerization, but rather form complexes with proteins.

Thank you for this suggestion. This sentence has been reworded in the revised manuscript (see lines 296-299).

  1. The conclusions are overly general and should provide more specific details about the key points discussed in each section of the review.

Thank you for this comment. The current version of the conclusion has been extended according to your comment. See section 8. (8. Conclusions).

  1. The article contains numerous typographical errors; the authors should thoroughly review the entire manuscript and make the necessary corrections.

Thank you for this suggestion. The manuscript has been thoroughly reviewed for typographical errors and corrected accordingly.
